# Unravelling the Role of Uncommon Hydrogen Bonds in Cyclodextrin Ferrociphenol Supramolecular Complexes: A Computational Modelling and Experimental Study

**DOI:** 10.3390/ijms241512288

**Published:** 2023-07-31

**Authors:** Pascal Pigeon, Feten Najlaoui, Michael James McGlinchey, Juan Sanz García, Gérard Jaouen, Stéphane Gibaud

**Affiliations:** 1Chimie ParisTech, PSL, 11 Rue Pierre et Marie Curie, CEDEX 05, 75231 Paris, France; 2Institut Parisien de Chimie Moléculaire (IPCM)—UMR 8232, Sorbonne Université, 4 Place Jussieu, 75252 Paris, France; 3EA 3452/CITHEFOR, Faculté de Pharmacie, Université de Lorraine, 5 Rue Albert Lebrun, 54000 Nancy, France; 4UCD School of Chemistry, University College Dublin Belfield, D04 V1W8 Dublin, Ireland; 51MSME, Université Gustave Eiffel, CNRS, 5 Boulevard Descartes, 77454 Marne-la-Vallée, France; juan.sanz.garc@gmail.com

**Keywords:** cyclodextrin, ferrocene, modelling, phase solubility, database, atypical hydrogen bond

## Abstract

We sought to determine the cyclodextrins (CDs) best suited to solubilize a patented succinimido-ferrocidiphenol (SuccFerr), a compound from the ferrociphenol family having powerful anticancer activity but low water solubility. Phase solubility experiments and computational modelling were carried out on various CDs. For the latter, several CD-SuccFerr complexes were built starting from combinations of one or two CD(s) where the methylation of CD oxygen atoms was systematically changed to end up with a database of ca. 13 k models. Modelling and phase solubility experiments seem to indicate the predominance of supramolecular assemblies of SuccFerr with two CDs and the superiority of randomly methylated β-cyclodextrins (RAMEβCDs). In addition, modelling shows that there are several competing combinations of inserted moieties of SuccFerr. Furthermore, the models show that ferrocene can contribute to high stabilization by making atypical hydrogen bonds between Fe and the hydroxyl groups of CDs (single bond with one OH or clamp with two OH of the same glucose unit).

## 1. Introduction

### 1.1. Cyclodextrins

Cyclodextrins (CDs) are cyclic oligosaccharides formed by glucose units bound together. Nowadays, they are used as excipients to improve the solubility of lipophilic compounds. Typical cyclodextrins contain a number of glucose units ranging from six to eight units in a ring and can be classified as α-cyclodextrin with six glucose subunits, β-cyclodextrin with seven glucose subunits (Figure 1) and γ-cyclodextrin with eight glucose subunits. Cyclodextrins are frequently transformed by chemical modification of the hydroxyl group, for example, to form alkyloxy derivatives (O-Me, O-Et, etc.) [1].

The most commercially viable strategies lead to the synthesis of partially and randomly alkylated cyclodextrins [2,3]. Chemically modified cyclodextrin derivatives available on the market are mixtures with different degrees of substitution [4]. These cyclodextrins can be used as excipients in many pharmaceutical formulations (e.g., tablets, aqueous parenteral solutions, eye drop solutions and nasal sprays) [5,6,7,8]. Although these randomly substituted cyclodextrins are widely used as solubilizers, they can form a wide range of complexes, making the characterization of these mixtures quite challenging. In silico models of these mixtures are usually focused on one or very few well-defined structures.

The term “well-defined” implies that all glucose units are identical. Figure 1 shows an example, 2-Me-βCD, in which all 2-positions (glucose numbering is not indicated) are methylated, and all 3- and 6-positions are unmethylated. CDs having at least two different glucose units are designated as undefined CDs in this work.

Although chemical modification of cyclodextrins might seem quite simple, it is in fact particularly challenging due to the presence of –OH groups in large numbers. More complex methods for selective modifications of cyclodextrins have been reviewed [9] and are currently used to prepare very specific varieties of them, such as the heptakis(2,3,6-tri-O-methyl)-β-cyclodextrin or the heptakis-(2,6-di-O-methyl)-β-cyclodextrin.

These cyclodextrins are much more expensive than the commercially available randomly methylated cyclodextrins (RAMEβCDs), making their use very restricted. In fact, researchers are often led to focus on cheaper, randomly substituted cyclodextrins for animal studies and for clinical trials. A non-trivial question arises when modelling cyclodextrins: what is the difference between solutions prepared with a well-defined cyclodextrin and with a mixture of randomly substituted cyclodextrins? We already know that the association constant of randomly methylated cyclodextrins depends greatly on their degree of methylation [10].

### 1.2. Ferrociphenol SuccFerr

In this paper, we report that we have performed a series of experiments for formulation with CDs on a ferrocidiphenol derivative (ferrociphenol with an additional *p*-hydroxyphenyl group: *N*-{4-ferrocenyl-5,5-bis-(4-hydroxy-phenyl)-pent-4-enyl}succinimide; SuccFerr (or P722), Figure 2) recently developed in our research group. SuccFerr is the result of an optimization of the ferrociphenol family [11] whose members act by apoptosis or senescence [12], where the pharmacophore is the *trans* (ferrocene-double bond-*p*-phenol) motif [13] (in blue in Figure 2), and the main metabolite is a quinone-methide [14,15] able to attack proteins [16,17]. However, the attachment of a succinimidyl group decreased the IC_50_ to around 40 nM on MDA-MB-231, A2780, A2780-Cis and K562 cell lines [18] thanks to an intramolecular stabilization [19]. Moreover, recent research on glioblastomas has shown that SuccFerr has a surprising selectivity on patient-derived cell lines (PDCLs) [20]. This patented compound [21] is currently being tested for its anticancer activity, but a suitable formulation was needed to achieve a usable formulation because of its lipophilicity and poor solubility. SuccFerr was recently formulated with various nanocapsules in its diphenol form, or to increase its lipophilicity and the load in lipid nanocapsules, as monoacetate [22] or diacetate [22,23,24], with or without the unexpected formation of a gel [22]. It was recently formulated with randomly methylated cyclodextrins (RAMEβCDs) and tested in vivo without noticeable adverse toxicity [25], but this last complexation has yet to be analyzed more finely by modelling and laboratory experiments.

## 2. Results

### 2.1. Experiments Prior to Modelling (Proof of Concept)

In our study, it was important to compare the modelling results with the laboratory solubilization tests. The complexing capacity is known to be very variable depending on the type of cyclodextrin, and it was therefore necessary to compare the results of solubilization of SuccFerr with various CDs.

The phase–solubility study was first described by Higuchi and Connors [26] to test the dissolving properties of the complexation.

In our experiment (Figure 3A), α-CD did not improve the solubility of SuccFerr, probably due to its small cavity size (internal diameter: 0.57 nm). In contrast, the methylated β-cyclodextrin (RAMEβCD) showed significant solubilization properties already previously described for the phthalimide compound [27]. The internal diameter of this CD is 0.68 nm, and it is well adapted to complex aromatic and heterocyclic moieties [28].

The phase solubility diagram of SuccFerr in the presence of RAMEβCD (see Figure 3A) exhibits a positive curvature, described as Ap-type by Higuchi and Connors [26]. This is observed if more than one CD can complex the drug corresponding to 1:2, 1:3, 1:4 (or more) stoichiometries. Our results confirm that RAMEβCD can be used to prepare a solution of SuccFerr.

It is noteworthy that DMβCD does not similarly improve the solubility as RAMEβCD does.

A second study was carried out to better understand the complexation mechanism. This experiment was performed to confirm the inclusion of the succinimidyl group in RAMEβCD. The spectrum of pure SuccFerr exhibits an absorption maximum at 300 nm corresponding to the succinimidyl moiety (Figure 3B). The ferrocenyl moiety is known to exhibit a λ_max_ at approximately 200 nm, whereas the λ_max_ of phenols is found at approximately 270 nm. The addition of RAMEβCD induces a hyperchromic effect at 300 nm (Figure 3B); in contrast, the inclusion of phenols and ferrocenyl moiety did not modify the spectrum at 254 nm and 270 nm.

The specific effect observed at 300 nm can be used to study the inclusion of the succinimidyl group using the method previously described for the phthalimide derivative [27] and derived from the Benesi–Hildebrand method. Consequently, we studied the variations in the absorption of SuccFerr as a function of the concentration of RAMEβCD using the spectroscopic data. Each absorption value was taken from our spectra at 300 nm.

The corresponding theoretical equation is:(1)$$\frac{\left[SuccFerr\right]}{∆A}=\frac{1}{E_{c}.l.{[CD}_{0}].K_{a}}+\frac{1}{E_{c}.l}$$
where
$$∆A=\frac{1}{\sum_{n}^{i=1}\frac{1}{∆A_{i}}}K_{a}=\frac{1}{\sum_{n}^{i=1}\frac{1}{K_{i}}}$$

This equation is similar to the Benesi–Hildebrand equations and is linear for [SuccFerr]/∆A as a function of 1/[CD_0_] (Figure 3C). Consequently, we can confirm that the linear curves obtained for double reciprocal curves of Benesi–Hildebrand correspond to a 1:1 apparent complexation with the succinimidyl group. In this equation, the absorbance coefficient and K_a_ can be defined as the harmonic mean of all the complexes if there is an inclusion of the succinimidyl moiety (see [27] for more details). [SuccFerr] is the concentration of SuccFerr, [CD0] is the initial concentration of free cyclodextrin, ε_c_ is the change in the molar absorption coefficient after complexation, and l is the path length.

The curve obtained from experimental data (i.e., [SuccFerr]/∆A as a function of 1/[CD_0_]) is presented in Figure 3C, and K_a_ was calculated (K_a_ = 50.618 ± 5.280 M^−1^).

### 2.2. Modelling and Web Application

We used modelling (semiempirical PM3 quantum-mechanical method) to determinate the type of association SuccFerr could make with CDs. Two main types of series have been produced. For the inclusion of SuccFerr into a single CD, eight possible systems were created (four moieties (Fc, Ph1, Ph2 and Succ, Figure 2) inserted by the narrow or the wide side of the CD, denoted 1-CD series S1 to S8). For the models with two CDs, only four series were made (2-CD series S1 to S4). These 12 trees have the same treatment in common. First, taking 1-CD models as an example, we calculated all the possible combinations of insertion (four moieties for SuccFerr × two insertion sides per CD × six possible well-defined CDs, see Appendix A for 2-CD). Then, for each of the eight (moiety–CD side) combinations, we selected the model with the most negative ∆E (the energy difference between the supramolecular assembly and the separated molecules, while keeping the same geometry). The aim of this selection was to start with the lowest possible energy for each of these eight models that served as the G0 (generation 0) model for creating series/trees. Then, the methylations on all oxygen atoms (21 for 1-CD systems) were inverted, creating a G1 level with 21 models in the tree. The best model (lowest ∆E) was selected to create the G2 level of the tree in the same way, and so on.

We created a web application to handle the data of these 12 trees, which also provided statistics, but a need arose to obtain more data on our calculations. This is why a second program was created in C programming language. Its purpose was to verify the models (building error that would invalidate this work), to search for certain hydrogen bonds of interest, to count the average number of methyl groups around Fc and to generate the static webpages, copies of the dynamic webpages created by the web application. In the latter case, the goal of generating these static webpages was to dispense with the web application for simplified read-only consultation (to view a particular model or to navigate into the trees). These webpages can be downloaded as a compressed archive [29].

To facilitate a better understanding of this concept, the code of the PHP program, the code of the C program and the database (scheme and data) are provided in a data repository [30] (including the static webpages cited above). The web application (CDModelTree) was also referenced on Software Heritage via Hal [31] and is executable and testable (even with any other inserted molecule that readers of this article would like to test, with up to two CDs for inclusion, and without any need to adapt the code) using the XAMPP pack [32]. More details can be found in the Appendix A.

The results of the calculations of these 12 series/trees are reported in Table 1.

## 3. Discussion

These calculations were carried out to determine which CDs should be the best fit for each moiety of SuccFerr. More particularly, these calculations were meant to serve, among other things, to determine the best suitable average methylation rate of these CDs.

Data collected in Table 1 and the representation of the variation in ∆E for the unique route going from G0 to the best model (called “main route”, Appendix A) for each of the 12 series/trees (Figure 4) show the superiority of the 2-CD systems, even at the G0 level (i.e., with well-defined CDs), which is predictable because there are more stabilizing interactions. Among the G0 systems of 1-CD, series 7 (insertion of Succ via the narrow side), which was the best, is exceeded in these mutation trees, at G1 by series S8 (insertion of Succ from the wide side) and even at G6 by series 2 (insertion of Fc from the wide side). This shows that no conclusion can be made based on well-defined CDs, but also that undefined CDs can sometimes perform better, even though the RAMEβCD mixture also contains CDs that are less efficient than well-defined CDs. At any rate, this behavior of 1-CD S8 and S2 merits further study.

### 3.1. Observed Contribution of the Iron Atom to Stability

Although Table 1 and Figure 4 show a superiority of 2-CD over 1-CD (lower ∆E for G0, lower average ∆E, best lower ∆E of series), the 1-CD series S8 experiment shows very different ∆E from other 1-CD series and is in competition with the ∆E of the 2-CD. The webpage displaying the G0 model of 1-CD S8 and its first-generation descendants (G1) can be consulted to see the data (using the web application by typing its ID number (6990) into the form, or more simply displaying the static saved version “1-CD-webpage-ID-6990.html” file, available inside the repository [30]). In the list of the G1 descendants, experiment ID 7010 shows the strongest variation in ∆E (between itself and its parent, ∆∆E = −139 kJ/mol) of all the 8892 calculated 1-CD experiments during the demethylation of the oxygen atom O_20_. This brusque variation in ∆E can be seen in Figure 4 at G1. In addition, the generations derived from this experience all show a significant destabilization during the re-methylation of this atom (as in model ID 7169) but also of O_19_ (for example, ID 7017). These anomalies in the data can be easily detected by SQL queries ordering the experiments by their ‘deltadelta’ field (representing ∆∆E in the database, Note A12). It is worth mentioning that thanks to these anomalies in the data, two atypical hydrogen bonds between hydrogens atoms from O_19_ and O_20_ and the central ferrocene iron atom have been detected. This is characterized by the formation of a clamp (–O_19_–H … Fe … H–O_20_–), as can be seen with the best model of this series (Appendix A). This type of bond, but intramolecular and without a clamp, has been reported with NH groups by an XRD analysis of a single crystal [33]. However, to the best of our knowledge, this is the first time that it has been described by modelling in supramolecular systems involving ferrocene and a cyclodextrin forming intermolecular Fe-H hydrogen bonds. This phenomenon is responsible for this decrease in energy, but also for its increase when the clamp is broken by methylating O_19_ or O_20_, both of which belong to the same anhydroglucose unit (positions 2 and 3 in the glucose numbering system). Another clamp (–O_10_–H … Fe … H–O_11_–, Appendix A) was also discovered in the 1-CD series S2, again associated with a strong positive value of ‘deltadelta’ (∆∆E) when O_10_ or O_11_ is methylated (for example, ID 3743 for O_10_ and ID 3744 for O_11_, seen when displaying the webpage of their common parent, model ID 3738). Unlike 1CD S8, where two Fe-H bonds (clamp) are formed at the same time, which explains the steep decrease in ∆E, the formation of the clamp for 1CD S2 is done gradually, as can be seen in Figure 4 (slower variation of ∆E, due to different stages passing through weaker Fe-H bonds formations). Even though both O_10_ and O_11_ were non-methylated since G1, this clamp is only formed at G9 due to the favorable influence of the modification of methylations managed by the web application. In a related way, the greatest increases in ∆E are not all due to the direct breaking of the clamp, but often also by indirect cleavage caused by steric hindrance of the latter during the methylation of an oxygen atom close to it, as methylation of O_1_ for series S8 (ID 7011 and ID 7030, having ∆∆E > 176 kJ/mol). In 2-CD systems, this phenomenon also exists, but the connection is single, or of an asymmetrical clamp type (one strong/short and one weak/long), probably because of the greater steric effect. Analyzing the geometries of all the 2-CD models, thanks to the C program (that does the same for 1-CD models) reading each of the XYZ files, we can confirm that no strong symmetrical clamp exists with Fe in any of these models, even though some simple bonds were found and are important for stability (Note A13).

Analyzing the geometries of the full dataset (XYZ files, using the C program) proved to be extremely useful in understanding the effect of the atypical and classic hydrogen bonds on the stability of supramolecular assemblies. We have systematically checked, for all the models, the existence, or not, of seven possible intermolecular hydrogen bonds. The maximum distance was set to 2 Å, and the results can be found in the provided Hbonds-1CD.txt [34] and Hbonds-2CD.txt [35] report files for 1-CD and 2-CD models, respectively. For SuccFerr, the atoms involved in these bonds are the iron atom, the two phenols (-O- and -H independently) and the two carbonyls (=O) of the imide group. For CDs, these are the hydroxyls (-O- and -H independently), the ethers and the anomeric group (-O-). For each series, the models were separated into two groups (worse models or best models, Note 8). The comparison of the number of each type of hydrogen bond found in these two groups showed that the effect is important especially when the iron atom is involved, even more so when it is implicated into a clamp (asymmetrical clamps (one bond > 2 Å and one bond < 2 Å) have been ignored). For phenols, the clamps only exist for the 2-CD series, whereas clamps involving the C=O groups were detected for both the 1-CD and 2-CD series. However, these two types of clamps do not carry as much weight in stability as those involving Fe (Note A13). To explain why, in the 2-CD models, no strong clamp involving the iron atom was detected, we can compare the 1-CD S8 (Figure 5) and 2-CD S1 series (which corresponds to the 1-CD S8 series with an additional CD in which Fc is inserted, Figure 6). We see on the models that this second CD impedes the approach of iron to two OH groups of the CD including Succ, as in the 1-CD S8 series, and instead makes a single bond with the iron atom (Figure 6).

The lengths of the Fe-H bonds found with PM3 seeming surprisingly short (≈1.95 Å), the best 1-CD series 8 model was then computed in DFT (see computational details). The latter confirms the formation of this clamp with the iron atom (Figure 5), with bond lengths of 2.77 and 2.69 Å for O_19_-H…Fe and O_20_-H…Fe, respectively (the corresponding angles are 159° and 166°). The hydrogen bond acceptors’ and donors’ distances are 3.69 and 3.64 Å for O_19_,Fe and O_20_,Fe. The calculation of the ferrocene + methanol models in water confirms the presence of Fe-H bonds in DFT in aqueous medium (in the compressed archive DFT.zip [36]). A further Natural Bond Orbitals (NBO) calculation has allowed us to estimate the stabilizing energy (E^(2)^, see Table 2) of the atypical H-bonds found between the Fe from the SuccFerr compound and the alcohols from the CD. Adding all the NBO (donor–acceptor) energy contributions (see Appendix A for further details) results in a total stabilisation energy of 86.45 kJ/mol (20.66 kcal/mol) and 119.24 kJ/mol (28.50 kcal/mol) for O_19_-H…Fe and O_20_-H…Fe, respectively. It is worth noting that in such particular supramolecular contacts, we observe not only a weak LP(Fe)-σ*(H_—_O) intermolecular interaction (3.80 kJ/mol) comparable to that which can be seen in common H-bond interactions, but we also observe σ(H_—_O)- LP*(Fe) stabilizing contributions, as well as quite uncommon strong (56.77 kJ/mol) LP*(Fe)- Rydberg (H) contributions among some other minor contributions.

The PM3 and DFT methods gave comparable results (see Table 3 for the lengths and angles of some hydrogen bonds for the best model ID 7759).

### 3.2. Contribution of Methyl Groups to Stability

One might think that since the ferrocenyl group is nonpolar, as are the phenyl groups, it could therefore interact better with methyl groups and prefer highly methylated CDs, or at least have many proximate methyl groups, as postulated by Buriez [37]. However, this is not reflected in our modelling for Fc, because the number of neighboring methyl groups is generally rather low, as shown in Figure 5. To understand the effect of the Fc-Me proximity on the stability, a systematic analysis has been performed by the C program (see code ‘NbMeFc’ in Table 1 and notes A7, A8, A9). We can see that the average number of these proximities are quite limited, and that there is no clear difference between best and worse models (except for 1CD S8 and S2 series, where some methyls should be removed to form the beneficial clamp with the iron).

Finding the best blend would not be of interest, as statistically, it only exists in small amounts in the RAMEβCD mix and is closely followed by many other assemblies in terms of stability; moreover, it would be idealistic to seek to synthesize the complex corresponding CDs, merely to dissolve SuccFerr better. Thus, it is better to consider macroscopic values, such as the average degree of methylation of the ideal mixture. By “ideal mix”, we mean that one should include in this calculation, for each series, only CDs participating in improved assemblies, that is, those for which ∆E < ∆EG0. These average states of methylation are calculated by the PHP program and are given in Table 1. However, relying only on these average methylations is restrictive, as it is found that the ideal range for each run (lowest and highest methylation rate of all improved CDs in any given run) is large. We can therefore deduce that SuccFerr can adapt to RAMEβCDs with very various methylation rates. The most important parameter is not this rate, but the distribution of CDs (in particular, the isomers), a parameter impossible to control during the synthesis of RAMEβCD, or difficult to assess by analysis. In conclusion, it has been shown in this work that the solubility of SuccFerr with methylated CDs depends on many variables, and it is no simple task to determine the best methylation rate for the CDs.

## 4. Materials and Methods

### 4.1. Complexation Studies in Aqueous Phase

#### 4.1.1. Phase Solubility Studies of SuccFerr Complexation

SuccFerr complexation with various CDs was evaluated using the phase-solubility method [26]. A suspension of a large excess of SuccFerr (30 mg) in 2 mL of aqueous solutions of the appropriate CD (concentrations ranging from 0.125 to 160 mM, pH adjusted to 7) was stirred in screw-capped amber vials for 24 h on a rock-and-roller agitator at 25 °C. Preliminary experiments indicated that equilibrium was reached after this 24 h stirring period. Each suspension was then centrifuged at 9000 g for 10 min and diluted from 1/10 to 1/1000 with acetonitrile, and the amount of dissolved SuccFerr was assessed by HPLC at 286 nm. Phase solubility curves (i.e., solubility of SuccFerr as a function of the CD concentration) were drawn for each CD.

#### 4.1.2. UV–Vis Experiments—Benesi–Hildebrand Method

First, UV–vis spectra (Cary 50 spectrophotometer, Varian, Les Ulis, France) were obtained for a concentration of 2.5 × 10^−5^ M of SuccFerr and various concentrations of CDs (0 to 1.1 × 10^−5^ M). RAMEβCD or HPβCD were added, and the SuccFerr concentration was kept at 2.5 × 10^−5^ M. All experiments were performed in 1% (*v*/*v*) DMSO/water mixture (the first dissolution of SuccFerr was performed in pure DMSO) to avoid precipitation, and the measurement was taken after 10 min to account for the kinetics of complex formation. Then, the mole-ratio titration method was used to calculate the values of the apparent binding constant (K_a_) [38,39,40]. These experiments were performed at various wavelengths obtained from the spectra (low concentration of CD). Another series of experiments was performed with higher CD concentrations (2 × 10^−6^–25 × 10^−5^ M, or high concentration of CD) to confirm the binding constants.

### 4.2. Computer Methods

#### 4.2.1. Software

Our experiments generated a large amount of data. We chose to create our own web application using the XAMPP pack (web server + database + PHP, version 8.2.0 was used) [32], which allowed us to design the very specific scheme required to store the data generated by our algorithms. We used this system to store our 13,261 models of inclusion of SuccFerr into one or two CD(s). The web browser serves as an interface between the scientist and our web application (entering data into a web form or displaying it). The PHP application acts as an intermediary with the MariaDB database, using SQL queries (see some examples in Appendix B), to store or access the data or even to generate statistics automatically.

The C program (to retrieve from the data repository [30]) was edited, compiled and executed using Code::Blocks version 20.03 64 bit [41]. It analyzes the XYZ files (checking for errors), creates the static webpages and performs additional statistics. More details can be found in Appendix A.

#### 4.2.2. Computational Details

All the SuccFerr-β-cyclodextrins complexes were computed using the program Spartan14 (Wavefunction, Irvine, CA, USA), performing a molecular mechanics optimization with the Merck molecular force field (MMFF) and a subsequent minimization with the semiempirical PM3 quantum-mechanical method. Starting from the best PM3 optimized geometry, a minimization was carried out using the Polarizable Continuum Model (PCM) in water. Density Functional Theory (DFT) was employed to perform this calculation using the ωB97XD functional with the diffuse-augmented polarization valence-triple-*ζ* (6-311G(d,p)) basis set including a set of *p* polarization functions for the hydrogen atoms and a set of *d* polarization functions for the second-row elements. For the iron atom, the uncontracted triple-*ζ* quality LANL08 basis set with an effective core potential (including 10 core electrons) was used in these calculations. This calculation was performed with the Gaussian 16 quantum package (Gaussian Inc., Wallingford, Connecticut, USA). On top of these DFT calculations, further NBO analyses were performed using the NBO version 3 integrated in Gaussian [42].

The PM3 method was used to determine the affinity of SuccFerr with the CD (∆E, Equation (2) or Equation (3)):for 1-CD: ∆E = E_(SuccFerr+CD)_ − E_SuccFerr_ − E_CD1_(2)
for 2-CD: ∆E = E_(SuccFerr+CD1+CD2)_ − E_SuccFerr_ − E_CD1_ − E_CD2_(3)

The energy of each element was calculated (energy method, for the ground state, no modification of geometry that should be the same as in the assemblage). The heat of formation (in kJ/mol) of each element and of the assemblage were copied from Spartan and pasted into the form of the webpage of the web application that calculated ∆E, and this was saved in the database.

## 5. Conclusions

This work has shown that the RAMEβCD mixture seems more suited than well-defined CDs to form stable associations with SuccFerr, and that the most probable assemblages are SuccFerr systems with two CDs, probably with a certain proportion of 1-CD series S8. The quantification results agree with the modelling showing that there are 2-CD assemblies. The modelling also revealed the special behavior of ferrocene, which forms atypical hydrogen bonds (one or two in the form of a clamp) between its iron atom and the hydrogen atoms of the hydroxyl groups of CDs. This result is particularly important because it has been possible, for the first time, to the best of our knowledge, to model the atypical hydrogen bonds between CDs and an anticancer molecule based on nonpolar ferrocene. These interactions are, in fact, responsible for the high affinity observed between the CDs and this molecule. Methyl groups also provide stability or instability depending on their position. For example, two OH groups at position 2 and 3 of the same glucose unit are not a sufficient parameter to allow a clamp with Fe, since some Me and OH groups should be placed at the right places elsewhere (6-Me-βCD in Appendix A does not have this clamp despite having seven glucose units with OH in position 2 and 3). Our method using trees of modifications of methylation permitted us to reach these particular configurations, and the discovery of these atypical bonds is due to the proper functioning of the experimental web application and to the automatic analysis of the model files by the C program.

## Figures and Tables

**Figure 1 ijms-24-12288-f001:**
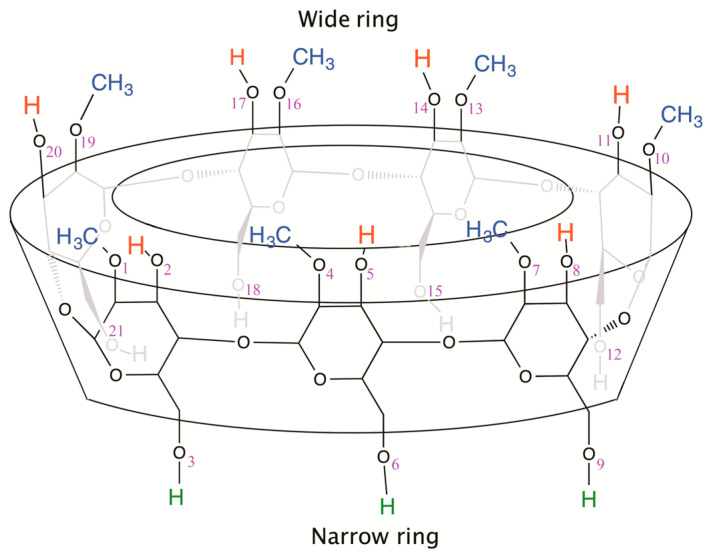
Representation of a β-cyclodextrin (CD, here 2-Me-βCD belonging to the category of well-defined CDs) with the oxygen atom numbering used for our modelling and database (glucose numbering is not displayed).

**Figure 2 ijms-24-12288-f002:**
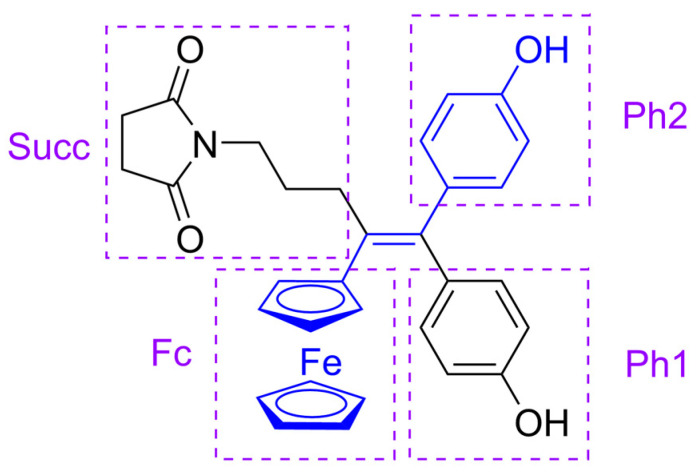
SuccFerr (the ferrociphenol motif is drawn in blue, and the moieties that can enter the cavity of the CDs are surrounded by purple rectangles with the abbreviations used in this work: the ferrocenyl group (Fc), phenol 1 (Ph1, *cis* to the Fc), phenol 2 (Ph2, *trans* to the Fc) and the succinimidylpropyl group (Succ).

**Figure 3 ijms-24-12288-f003:**
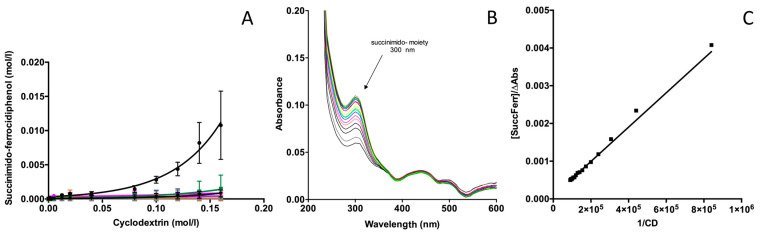
(**A**) Phase solubility diagram of SuccFerr in the presence of αCD (-■-), RAMEβCD (-●-), DMβCD (-▼-) HPβCD (-■-), HEβCD (-▲-), SBEβCD (-●-), γCD (-♦-). (**B**) Spectra of Succ:Ferr (2.5 × 10^−5^ M). The control (i.e., lower curve) was obtained with 1% DMSO. The other curves were obtained by addition of various amounts of RAMEβCD, exhibiting a hyperchromic effect at 300 nm. (**C**) Benesi–Hildebrand plots for the complexation of SuccFerr in RAMEβCD (λ = 300 nm).

**Figure 4 ijms-24-12288-f004:**
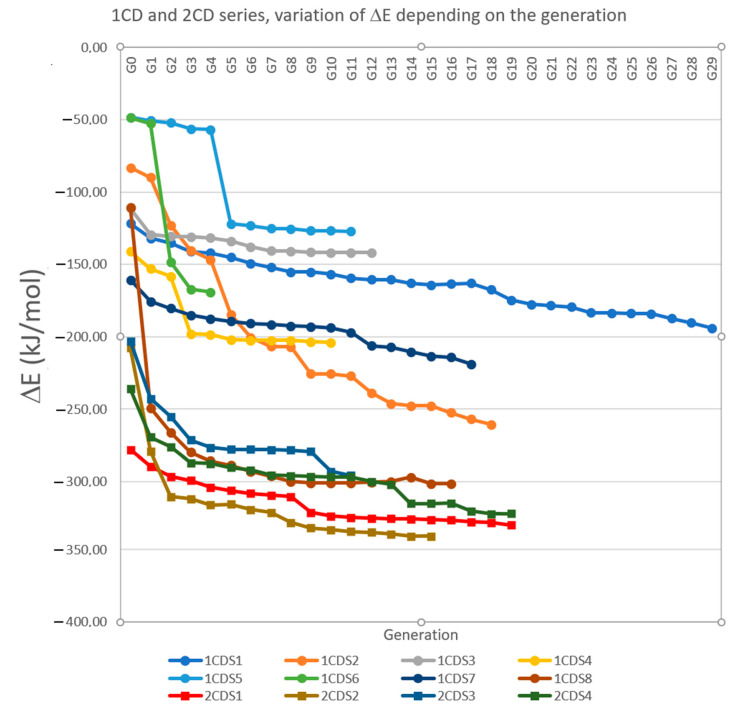
For the eight 1CD series (S1 to S8) and the four 2CD series (S1 to S4), unique route in the trees, starting from G0 and ending when reaching the best model of the series with display of the variation in ∆E (kJ/mol) at each generation. Some values can be found in Table 1: the starting ∆E (∆E G0), the ending ∆E (∆E of the best model: best ∆E), and the generation of the best model (final generation: Gbest). The PHP code to retrieve the data needed to fill this graphic from the database is explained in Appendix A.

**Figure 5 ijms-24-12288-f005:**
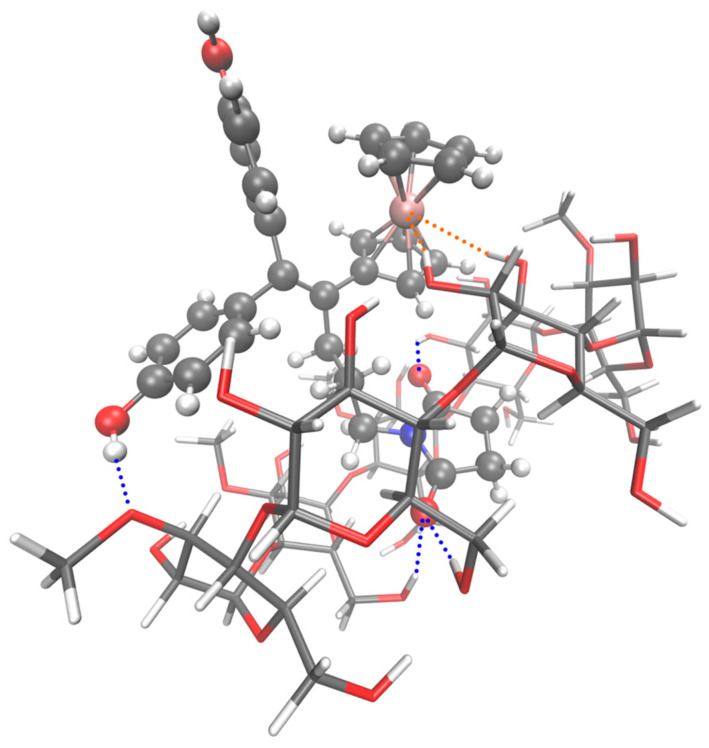
Molecular model of the best 1-CD system (series S8, ID 7759, Succ entering by the wide side of the CD) computed with Gaussian 16 at the DFT level of theory. SuccFerr is represented with a “ball-and-stick” model, and the CD is represented with a “licorice” model. Orange dashed lines (clamp between the iron atom and O_19_–H and O_20_–H) represent the two atypical hydrogen bonds between the iron atom and the CD. Another clamp between one of the C=O group and O_12_–H and O_18_–H is also visible, as bonds between the phenol group (Ph2) and O_14_–Me and between the second C=O of imide and O_5_-H (blue dashed lines).

**Figure 6 ijms-24-12288-f006:**
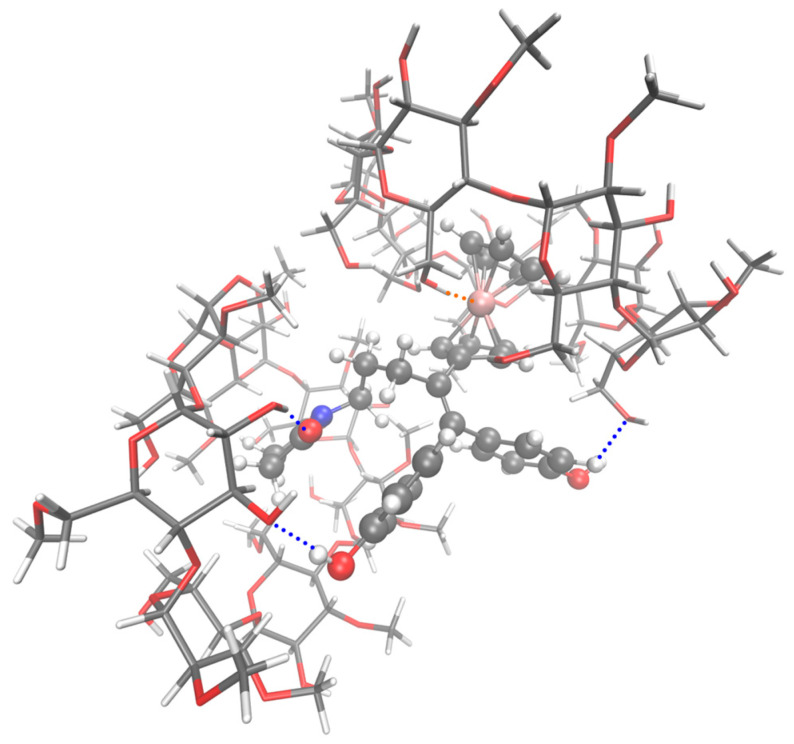
Molecular model of the best 2-CD series S1 system (ID 586, Succ entering by the wide side of CD1 and Fc entering by the narrow side of CD2) computed at the semiempirical PM3 level of theory. SuccFerr is represented with a “ball-and-stick” model, and CD1 and CD2 are represented with a “licorice” model. Orange dashed line is the hydrogen bond between the iron atom and O_36_–H of CD2.

**Table 1 ijms-24-12288-t001:** Comparison of all the 1-CD series and of four of the 24 possible 2-CD series (for each series/tree, average and best ∆E (in kJ/mol), generations reached, methylation rate (average Me per glucose unit), number of calculated samples and free CDs, …), calculated with the semiempirical PM3 quantum-mechanical method.

Series	1-CD S1	1-CD S2	1-CD S3	1-CD S4	1-CD S5	1-CD S6	1-CD S7	1-CD S8
Moiety-side ^A^	Fc NS	Fc WS	Ph1 NS	Ph1 WS	Ph2 NS	Ph2 WS	Succ NS	Succ WS
βCD G0 ^B^	2-Me	3-Me	2-Me	3,6-diMe	2-Me	6-Me	2,3-diMe	3,6-diMe
∆E G0 ^C^	−122	−83	−111	−141	−48	−49	−161	−111
Average ∆E ^D^	−154	−144	−127	−177	−68	−98	−187	−274
Best ∆E ^E^	−194	−261	−142	−204	−127	−169	−219	−302
Max G/Gbest ^F^	33/29	19/18	13/12	13/10	12/11	4/4	18/17	21/16
Average Me ^G^	1.4	1.07	1.09	1.73	1.04	1.31	1.34	0.98
Me Domain ^H^	0.6–2.4	0.3–2.0	0.6–1.6	1.1–2.3	0.6–1.3	1.0–1.6	0.6–2.3	0.3–2.1
Fe-H(CD) ^I^	191(0) 1(0)	497(118) 145(0)	0(0) 0(0)	60(0) 0(0)	0(0) 0(0)	1(0) 0(0)	929(0) 0(0)	3414(1664) 2(0)
NbMeFc ^J^	1.44 0.89	1.60 2.37	0.53 1.57	1.26 2.88	0.02 0.14	1.05 0.52	1.41 1.30	0.93 1.33
Samples ^K^	2726	1994	571	326	187	79	1106	1903
ID G0 ^L^	1	2727	4721	5292	5618	5805	5884	6990
Free CDs ^M^	2690	1877	550	305	166	57	1084	1872
**Series**	**2-CD S1**		**2-CD S2**		**2-CD S3**		**2-CD S4**	
Moiety-side 1 ^A^	Fc NS		Fc NS		Fc NS		Succ NS	
Moiety-side 2 ^A^	Succ WS		Ph2 WS		Ph1 WS		Ph1 WS	
βCD G0 ^B^	3-Me		3-Me		2,3-diMe		2-Me	
∆E G0 ^C^	−278		−208		−203		−236	
Average ∆E ^D^	−271		−293		−248		−283	
Best ∆E ^E^	−330		−338		−296		−326	
Max G/Gbest ^F^	22/19		15/15		12/11		27/26	
Average Me ^G^	1.33, 1.36		1.03, 1.16		1.94, 1.88		0.97, 0.81	
Me Domain ^H^	0.9–1.9		0.6–1.4		1.6–2.3		0.4–1.4	
Fe-H(CD) ^I^	790(0) 58(0)		534(0) 24(0)		367(0) 48(0)		1175(0) 3(0)	
NbMeFc ^J^	1.11 2.05		0.97 1.58		0.94 0.87		1.14 1.80	
Samples ^K^	1138		642		449		1229	
ID G0 ^L^	1		1139		1781		2230	
Free CDs ^M^	495		253		195		451	

These data were extracted from the webpages generated by the PHP program and from the report files generated by the C program that can be consulted in the data repository [30]. (A) Moiety of SuccFerr inserted inside the cavity of the CD and entrance side: NS = narrow side, WS = wide side. (B) Well-defined CD used as the G0 starting system. For 2-CD systems, the same CD was used twice. (C) ∆E of the starting system G0 (Note A1). These values are the best ones of Appendix A. (D) Average ∆E of all the experiments (Note A2). (E) Best ∆E reached by the tree (Note A3). (F) Max G = higher generation reached by the tree (Note A4), Gbest = generation of the best experiment (Note A5). (G) Average Me = average methylation per glucose unit for systems better than G0 (for each of the two CDs for the 2-CD series, see details in SI). (H) Methylation-per-glucose-unit domains for systems better than G0 (for the 2-CD series, union of the overlapping domains of the two CDs) (see details in SI). (I) Code ‘Fe-H(CD)’ indicates, for the whole series, the total number of hydrogen bonds between Fe and a hydrogen atom of a CD for the best (first row) and worse (second row) systems, respectively (Notes A6, A7, A8). The values inside parenthesis indicate the total number of clamps with Fe. (J) Code ‘NbMeFc’ indicates the average number of Me close to the ferrocene moiety for the best (first row) and worse (second row) systems, respectively (Notes A7, A8, A9). (K) Number of calculated experiments for each series (Note A10). (L) ID (model number) of the G0 system. (M) Total number of distinct CDs, when freed from the supramolecular assemblage (Note A11).

**Table 2 ijms-24-12288-t002:** Second-order perturbation stabilization energy contributions to the atypical H-bond Fe-HO interactions. NBO labels: LP = lone pair, LP* = antibonding lone pair, CR = core, σ* =Antibonding sigma bond.

NBO Donor	NBO Acceptor	E^(2)^ (kJ/mol)
**Fe ----- H_—_O_19_**		
LP_All_ (Fe)	Rydberg_All_ (H)	1.13
LP_All_ (Fe)	σ*_All_ (H_—_O)	3.80
LP*_All_ (Fe)	Rydberg_All_ (H)	56.77
LP*_All_ (Fe)	Rydberg_All_ (O)	9.23
LP*_All_ (Fe)	σ*_All_ (H_—_O)	4.48
σ_All_ (H_—_O)	LP*_All_ (Fe)	9.83
σ_All_ (H_—_O)	Rydberg_All_ (Fe)	0.29
LP_All_ (O)	LP*_All_ (Fe)	0.92
**Fe ----- H_—_O_20_**		
CR_All_ (Fe)	Rydberg_All_ (H)	0.21
LP_All_ (Fe)	Rydberg_All_ (H)	3.14
LP_All_ (Fe)	σ*_All_ (H_—_O)	2.90
LP*_All_ (Fe)	Rydberg_All_ (H)	98.40
LP*_All_ (Fe)	σ*_All_ (H_—_O)	4.43
σ_All_ (H_—_O)	LP*_All_ (Fe)	8.61
σ_All_ (H_—_O)	Rydberg_All_ (Fe)	0.50
LP_All_ (O)	LP*_All_ (Fe)	1.05

**Table 3 ijms-24-12288-t003:** Lengths and angles of some hydrogen bonds for model 2CD S8 ID 7759 calculated by PM3 and DFT.

Type of Hydrogen Bond	Length PM3 (Å)	Length DFT (Å)	Angle PM3 (°)	Angle DFT (°)
O_19_-H…Fe	1.933	2.774	167	159
O_20_-H…Fe	1.896	2.694	177	166
C=O(1)…H(CD)	1.910	2.010	148	128
C=O(2)…H(CD) (first)	1.838	1.864	166	164
C=O(2)…H(CD) (second)	1.858	2.087	162	163
H(Ph2)…O(CD)	1.843	1.751	167	173
H…O intra CD (1)	1.863	1.874	163	173
H…O intra CD (2)	1.825	1.815	168	175
H…O intra CD (3)	1.862	1.868	172	164
H…O intra CD (4)	1.816	1.836	174	175
H…O intra CD (5)	2.474	2.294	108	111
H…O intra CD (6)	1.862	1.868	172	164

O_19_-H…Fe: first hydrogen bond of the clamp with Fe, O_20_-H…Fe: second hydrogen bond of the clamp with Fe, C=O(1)…H(CD): hydrogen bond between the first CO of imide and H of the CD, C=O(2)…H(CD): hydrogen bond between the second CO of imide and H of the CD (first and second hydrogen bond of the clamp), H(Ph2)…O(CD): hydrogen bond between H of phenol 2 and O of CD, H…O intra CD (1 to 6): some selected intra-CD hydrogen bonds.

## Data Availability

A part of the data supporting this article is available within the article and Appendix A and more detailed descriptions of the program and examples of SQL queries adapted to our database). The main part of the data is publicly accessible in the ‘Recherche Data Gouv’ repository (space of Sorbonne Université) at https://doi.org/10.57745/CBUPP3 (accessed on 22 April 2023) [30]. This data repository contains the database in various formats (CSV, SQL (preferred to import the database [43]), JSON and XML), 13361 models in XYZ format, 28 original Spartan files, the S8 DFT geometry in XYZ format, the files created by analysis of the models by the C program, the C program itself and the PHP program. See the README.txt file for details [44]. The PHP program was also referenced on Software Heritage via Hal [31].

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
