# Peer review of "Unravelling the Role of Uncommon Hydrogen Bonds in Cyclodextrin Ferrociphenol Supramolecular Complexes: A Computational Modelling and Experimental Study"

_ijms, 2023, doi:10.3390/ijms241512288_

Round 1

Reviewer 1 Report (Previous Reviewer 1)

The authors have made many modifications to the paper, considering most of the referees' suggestions. In this form, I recommend publication in IJMS. 

Author Response

Thank you for accepting our manuscript.
Pascal Pigeon

Reviewer 2 Report (New Reviewer)

The authors of the submitted manuscript propose a computational approach to study complex formation between their prominent molecule SuccFerr and randomly methylated beta-CD. The chosen theoretical model is relevant to practical conditions of pharmaceutical formulations rather than to idealized complexes formed by well-defined host and guest molecules, so the scientific merit of this work is not in doubt.

From my side, I'm totally unable to judge the IT aspects and tree generation algorithms, but I would like to point out some possible improvements of data acquisition and presentation.

First of all, I think that the application of the semiempirical quantum chemical PM6-DH2 method, extended by dispersion and H-bond correction terms, instead of PM3 may be more appropriate in general, as it reliably describes various types of noncovalent complexes (J. Chem. Theory Comp. 2009, 5, 1749-1760). As far as I remember, the SPARTAN'14 package does not have an implementation of this method, but it can be freely used in MOPAC. Probably it's too late to ask for new calculations, as the authors are already struggling with the vibrational analysis required in the previous revision step, but it can be a good hint for further calculations.

In this context I miss the summarized correlations between DTF data and PM3 calculations, the comparison is given in a rather scattered manner. While the chosen computationally expensive wB97XD/6-311G(d,p) DFT method includes the dispersion corrections necessary to properly treat the noncovalent interactions, PM3 is a very rough but computationally inexpensive level of theory. However, once the results of these calculations show satisfactory correlations, it can justify the use of PM3 for further studies.

Table 1 contains a lot of numerical data, which is somewhat difficult to understand, as one has to follow the explanations for almost every line. Probably the authors can try to combine the data into some diagrams showing the distributions of energies and relevant parameters depending on different generations.

The authors demonstrate a superiority of the 2-CD over the 1-CD binding mode (line 227), but no graphical visualizations of such assemblies with identification of relevant intermolecular interactions are provided either in the manuscript or in the supporting information.

Finally, the detection of hydrogen bonds involving the iron atom based only on the proximity of certain atoms may not be sufficient to draw definitive conclusions about their role in the stabilization of certain complexes. I think that the wavefunctions obtained by DFT calculations can be further analyzed using the QTAIM formalism to identify the critical points of the bonds corresponding to the attractive interactions, i.e. H-bonding, with further estimation of the energies of these bonds using Espinosa-Lecomte-Mollins correlation schemes (Chem. Phys. Lett., 1999, 300, 745-748. Chem. Phys. Lett., 1998, 285, 170-173). Multiwfn package can be of great help in this context.

Author Response

Dr Pascal Pigeon

Sorbonne Université

Paris

Dear Professors Miguel A. Esteso and Carmen M. Romero,

I am pleased to resubmit our corrected article "Unravelling the role of uncommon hydrogen bonds in cyclodextrin ferrociphenol supramolecular complexes: a computational modelling and experimental study" to International Journal of Molecular Sciences as part of the special issue "Cyclodextrins: Properties and Applications".

This work deals with the solubilization with cyclodextrins of our molecule "SuccFerr", an experimental organometallic anti-cancer drug with low water solubility.

We hope you will find our corrected article suitable for publication in the journal and this special issue. We tried our best to improve this article with the advices of the reviewers, in particular reviewer 2 that gave very good advices for the second review.

  • We confirm that neither the manuscript nor any parts of its content are currently under consideration or published in another journal.
  • All authors have approved the manuscript and agree with its submission to (International Journal of Molecular Sciences).

Best regards

Dr. Pascal Pigeon

Sorbonne Université

Reviewer 2:

The authors of the submitted manuscript propose a computational approach to study complex formation between their prominent molecule SuccFerr and randomly methylated beta-CD. The chosen theoretical model is relevant to practical conditions of pharmaceutical formulations rather than to idealized complexes formed by well-defined host and guest molecules, so the scientific merit of this work is not in doubt.

From my side, I'm totally unable to judge the IT aspects and tree generation algorithms, but I would like to point out some possible improvements of data acquisition and presentation.

First of all, I think that the application of the semiempirical quantum chemical PM6-DH2 method, extended by dispersion and H-bond correction terms, instead of PM3 may be more appropriate in general, as it reliably describes various types of noncovalent complexes (J. Chem. Theory Comp. 2009, 5, 1749-1760). As far as I remember, the SPARTAN'14 package does not have an implementation of this method, but it can be freely used in MOPAC. Probably it's too late to ask for new calculations, as the authors are already struggling with the vibrational analysis required in the previous revision step, but it can be a good hint for further calculations.

The use of MOPAC (free) is a good idea to overcome the limitations of Spartan (paid). This method was not used here and as the reviewer stated, it is too late to recalculate more than 13,000 models using it. We thank the referee for the suggestion for future calculations.

In this context I miss the summarized correlations between DTF data and PM3 calculations, the comparison is given in a rather scattered manner. While the chosen computationally expensive wB97XD/6-311G(d,p) DFT method includes the dispersion corrections necessary to properly treat the noncovalent interactions, PM3 is a very rough but computationally inexpensive level of theory. However, once the results of these calculations show satisfactory correlations, it can justify the use of PM3 for further studies.

We added Table 3 to compare some lengths and angles for the best 1CD model calculated with PM3 and DFT and we can see that PM3 could be an interesting alternative to the time-consuming DFT. But DFT should be used to confirm the PM3 calculations for some selected models. For now, with the calculation power of computers, it is not possible to use DFT as routine method to calculated our trees of modification of methylation.

Table 1 contains a lot of numerical data, which is somewhat difficult to understand, as one has to follow the explanations for almost every line. Probably the authors can try to combine the data into some diagrams showing the distributions of energies and relevant parameters depending on different generations.

To better see the series, Figure 4 has been added. It allows to see the evolution of the series with the generations. Moreover, it also shows that 2CD models are better than 1CD models, except 1CD S8 series. This figure makes it possible to better understand what is described in the text (better than table 1, even though all the data inside this table 1 cannot be described by this figure) and we thank the reviewer for this idea which improves our description.

The authors demonstrate a superiority of the 2-CD over the 1-CD binding mode (line 227), but no graphical visualizations of such assemblies with identification of relevant intermolecular interactions are provided either in the manuscript or in the supporting information.

This is right. Figure 6 has been added and it describes the best model of 2CD series 1 calculated in PM3 (taken as an example for representing 2CD series). The reason we omitted this figure was because we focused on DFT for the Figures inside the article (and PM3 in SI for 1CD models, not for 2CD models though). However, DFT for 1CD models is very time-consuming, and with 2 CDs, it was absolutely unthinkable for us (especially on Spartan14!). We can see the simple hydrogen bond with Fe, as some hydrogen bonds with the two phenols and also a CO of imide group.

Finally, the detection of hydrogen bonds involving the iron atom based only on the proximity of certain atoms may not be sufficient to draw definitive conclusions about their role in the stabilization of certain complexes. I think that the wavefunctions obtained by DFT calculations can be further analyzed using the QTAIM formalism to identify the critical points of the bonds corresponding to the attractive interactions, i.e. H-bonding, with further estimation of the energies of these bonds using Espinosa-Lecomte-Mollins correlation schemes (Chem. Phys. Lett., 1999, 300, 745-748. Chem. Phys. Lett., 1998, 285, 170-173). Multiwfn package can be of great help in this context.

We would really like to thank the referee for suggesting to go further in the analysis of the energy estimation for the atypical H-bond interactions that we describe in this paper. As a matter of fact, such analysis strengthens the message of the manuscript. However, it is well known that AIM theory presents its limitations when it comes to the description of weak interactions (in particular H-bonds, see ref. Lane, J. R., Contreras-García, J., Piquemal, J.-P., Miller, B. J., & Kjaergaard, H. G. (2013). Are Bond Critical Points Really Critical for Hydrogen Bonding? Journal of Chemical Theory and Computation, 9(8), 3263–3266. doi:10.1021/ct400420r). Indeed, the absence of bond critical points (BCPs) does not necessarily mean that there are no H-bonds. Fortunately, we find two critical points between the Fe and the HO- groups involved in the atypical H-bond interactions described, however, further analysis on the bond paths (which do not mean necessarily correlate with chemical bonds or weak interactions, see ref. J. Phys. Chem. A 2009, 113, 38, 10391–10396. doi.org/10.1021/jp906341r) is also required and needs to be done very carefully, taking into account the AIM limitations due to its locality. There are alternatives to the BCPs and bond paths such as the Non-Covalent Interactions (NCI) index (which shows an attractive interaction between the Fe and the HO- groups involved). However, we think that such detailed and refined analysis of these not so-well described atypical interactions using AIM theory (and/or alternatives electron density descriptors) deserves a further and detailed study on its own and it is not really the scope of this manuscript. Furthermore, the formulas proposed by Espinosa-Lecomte-Mollins as well as a recent improvement of it for the estimation of H-bond energies (J. Comput. Chem., 40, 2868 (2019) DOI: 10.1002/jcc.26068) present a limitation due to the nature of the datasets use to derive those equations. In fact, all the molecules used in those studies are regular organic molecules with “classical” H-bonds which do not represent the organometallic complexes that we are studying in this manuscript, thus, the equations may not be valid for our system.

On the other hand, an alternative analysis to further estimate the atypical H-bond interaction energies has been used. We have used the Natural Bond Orbital (NBO) analysis to estimate the H-bond interaction energies. The results of these analyses have been included in the text strengthening its message.

Round 2

Reviewer 2 Report (New Reviewer)

The authors have improved the presentation of the results and this interesting work can be accepted in its present form!

For further studies I can recommend to switch to a more powerful package besides MOPAC, namely ORCA, which is also free and has perfect parallelization algorithms with many DFT options for fast and accurate analysis of geometry, electronic structure, spectral properties, etc.

I also think that using Grimme's 3c methods, such as r2SCAN-3c, could be more advantageous in terms of computational time for large systems compared to SPARTAN, whose capabilities are too limited for software costing several thousand dollars.

I wish the authors good luck in their further work!

This manuscript is a resubmission of an earlier submission. The following is a list of the peer review reports and author responses from that submission.

Round 1

Reviewer 1 Report

In this work, the authors related the modeling of new cyclodextrins aiming to design an inclusion complex that could overcome the low water solubility of a ferrociphenol derivative. To do this, they build up a sophisticated database of structural models. This is a critical perspective regarding supramolecular interactions since intricated, and multiple equilibria are expected. The work has merit, but I still have comments that can help improve it and make it more suitable for a broader audience.

General concerns:

The sentences: “In silico models of these mixtures are usually focused on one or very few well-defined structures” (page 2, line 47); and “In some cases, randomly substituted cyclodextrins showed better dissolution properties” (page 2, line 69) needs at least one reference confirming it, mainly because they are critical to building up the author’s hypothesis.

About the theoretical methods, It is worth mentioning that Semi-empirical is known to keep the overall symmetry of large systems, which can be a good choice when the starting geometry is given from x-ray data, for example. However, the authors proceed with a pre-minimization step with molecular mechanics before the PM3 calculation. How could this impact the geometries and also the energetics contributions?

Authors cite that they use DFT in water as implicit solvation within Spartan. The basis set and the solvent model applied are depicted in the Figure 4 caption. However, why they use a basis set without polarization needs to be clarified. How could this impact the description of all intermolecular interactions, especially the Iron-Hydrogen bond?

About the PM3 and DFT methods, it is also worth mentioning that vibrational analysis was not reported. Since these systems have great structural flexibility, some note should be made about the nature of these stationary points on the PES.

Authors could dedicate a few sentences to discuss thermodynamics contributions to evaluate inclusion complexes. It is well known that entropic and enthalpic balance can bring different Gibbs energy variations upon inclusion. So, is it entirely fair to use just DH to evaluate this?

Other notes:

At SI, some symbols are missing

Tables S1, S2, and S3 should have energy unities in their captions.

The authors describe the hydrogen bonds' distances, but how about their angles and estimated energies? And also, how close are these hydrogen bond acceptors and donors to each other?

Reviewer 2 Report

 The main question addressed by the research consists in finding the cyclodextrins (CDs) which are best suited to solubilize a patented succinimido-ferrocidiphenol (SuccFerr/P722).  The Special Issue "Cyclodextrins: Properties and Applications" is the most eligible match for this work, and there is no doubt in relevancy to the field of macromolecular chemistry. The topic is indeed can be considered as original, as it represents a logical continuation of the work started by the authors earlier; the ferrocidiphenol derivative SuccFerr has recently been developed in their research group, and now they carry out a series of experiments on the solubility of this compound in different types of cyclodextrins, because it has very low solubility in water. In this respect, one can say that the topic addresses a specific gap. I’m inclined to recommend this work for publishing in IJSM, though, some aspects persuade me to inquire some major improvements.

First of all, my main concern is how the text is written. Redundancy and unclearness of narration are the main disadvantages of this article. For example, in the introductory section there is a subsection “In-house developed software to analyze the produced scientific data,” where the authors describe free pack XAMPP that can be used to consult a very large scientific database, as well as they mention some other distinguished technologies that can help in the exploitation and management of large amounts of computational data. This is great, but what is the purport of such an excerpt in the end of introduction? Did the authors try to say that they have used these packages in their work (alas, this vaguely becomes clearer only in the subsection 2.2 of the “Results”)? Why this technical information is included into the introduction? As to my humble opinion, it is the “Methods and Materials” section, where such technical excerpts ought to be placed. Another example is the text in lines 79-93, which, apparently, should be removed or rewritten. This perplexed and unrelated paragraph presents a very specific information on biological activity of ferrocene-double-bond-p-phenol motif, which stray away the reader from the main line of the work. Unfortunately, the whole text is superfluous and full of inconsistences. I frequently see that the following up sentence is not logically connected with the previous one – the narration is lacking logic. On that occasion, I recommend to rewrite the manuscript so as to make it readable and understandable. It is also very desirable to significantly reduce the text by moving secondary technical information to Supplementary materials.

            I also have several specific questions:

1.      Please, describe in more detail the equation 1, representing the absorption value of SuccFerr as a function of the concentration of RAMEβCD. Each parameter of this equation should be referred to in the text. What do the dots entail? Is it a faulty attempt to depict a multiplication sign or is it an uncommon designation of whatnot mathematical operator? Please provide physical explanation of what you have observed in experiment – the linearity of the absorption function.

2.      The PM3 results can safely be removed from this work as they do not represent any value.

3.      The 6-31G* basis set is so small of a size that it is hardly applicable to appropriately describe so an intricate electronic system as that of the considered complexes. I suggest to recalculate all in 6-311G** or in cc-pVTZ basis set. The computational details (method/basis set) should be placed in the text, not in the captions to figures.

4.      As to the minor remarks, I suggest to pay attention to the font in which the Greek letter “β” is printed in the RAMEβCDs abbreviation throughout the text - it is strangely looking and different from that used in line 25 (in abstract).

Not so bad.